# Comparative Genomics of DH5α-Inhibiting *Escherichia coli* Isolates from Feces of Healthy Individuals Reveals Common Co-Occurrence of Bacteriocin Genes with Virulence Factors and Antibiotic Resistance Genes

**DOI:** 10.3390/antibiotics14090860

**Published:** 2025-08-26

**Authors:** Shuan Er, Yichen Ding, Linda Wei Lin Tan, Yik Ying Teo, Niranjan Nagarajan, Henning Seedorf

**Affiliations:** 1Temasek Life Sciences Laboratory, 1 Research Link, Singapore 117604, Singapore; shuan_er@yahoo.com.sg (S.E.); yichending@hotmail.com (Y.D.); 2Saw Swee Hock School of Public Health, National University of Singapore, 12 Science Drive 2, Singapore 117549, Singapore; linda_tan@nus.edu.sg (L.W.L.T.); ephtyy@nus.edu.sg (Y.Y.T.); 3Department of Statistics and Data Science, National University of Singapore, Singapore 117546, Singapore; 4Genome Institute of Singapore, A*STAR, Singapore 138672, Singapore; nagarajann@gis.a-star.edu.sg; 5Yong Loo Lin School of Medicine, National University of Singapore, Singapore 117597, Singapore; 6Department of Biological Sciences, National University of Singapore, Singapore 117558, Singapore

**Keywords:** bacteriocin, antimicrobial resistance, multi-drug resistance, extraintestinal pathogenic *E. coli* (ExPEC), gut microbiome

## Abstract

**Background/Objectives**: The presence of multi-drug-resistant (MDR) bacteria in healthy individuals poses a significant public health concern, as these strains may contribute to or even facilitate the dissemination of antibiotic resistance genes (ARGs) and virulence factors. In this study, we investigated the genomic features of antimicrobial-producing *Escherichia coli* strains from the gut microbiota of healthy individuals in Singapore. **Methods**: Using a large-scale screening approach, we analyzed 3107 *E. coli* isolates from 109 fecal samples for inhibitory activity against *E. coli* DH5α and performed whole-genome sequencing on 37 representative isolates. **Results**: Our findings reveal genetically diverse strains, with isolates belonging to five phylogroups (A, B1, B2, D, and F) and 23 unique sequence types (STs). Bacteriocin gene clusters were widespread (92% of isolates carried one or more bacteriocin gene clusters), with colicins and microcins dominating the profiles. Notably, we identified an *hcp-et3-4* gene cluster encoding an effector linked to a Type VI secretion system. Approximately 40% of the sequenced isolates were MDR, with resistance for up to eight antibiotic classes in one strain (strain D96). Plasmids were the primary vehicles for ARG dissemination, but chromosomal resistance determinants were also detected. Additionally, over 55% of isolates were classified as potential extraintestinal pathogenic *E. coli* (ExPEC), raising concerns about their potential pathogenicity outside the intestinal tract. **Conclusions**: Our study highlights the co-occurrence of bacteriocin genes, ARGs, and virulence genes in gut-residing *E. coli*, underscoring their potential role in shaping microbial dynamics and antibiotic resistance. While bacteriocin-producing strains show potential as probiotic alternatives, careful assessment of their safety and genetic stability is necessary for therapeutic applications.

## 1. Introduction

The emergence of multi-drug-resistant (MDR) microorganisms is posing serious threats to global health care and raising calls for novel approaches to prevent the spread of infectious diseases. The rising prevalence of MDR strains in healthy subjects is, in this regard, particularly worrisome, as such individuals could contribute to the dissemination of resistance genes into populations and/or into pathogenic bacteria. In previous studies, we reported a high abundance of a multi-drug-resistant *E. coli* strain, 94EC, in the feces of a healthy subject that had not been treated with antibiotics prior to sampling [1,2]. This strain harbored resistance genes for seven different classes of antibiotics, including last-line antibiotics, such as tigecycline and colistin [3]. It was noted that 94EC colonies displayed antimicrobial activity against other Enterobacteriaceae strains on agar plates (previously unpublished). The underlying causes for the antimicrobial activity and high abundance in the host remained unclear, and whole-genome sequencing of the strain was therefore performed in order to analyze the genome for potential contributing features.

Genome analysis revealed the presence of several bacteriocin gene clusters within the strain (previously unpublished). Bacteriocins are known to have important roles in ecology and microbial population dynamics and have also been of interest for applications in biotechnological processes or as potential alternatives for antibiotic therapeutics [4,5,6]. However, their co-occurrence with antibiotic resistance genes and their role in the spread of antibiotic resistance or in stabilizing the resistome are less well understood. Such combinations may influence microbial fitness, strain persistence, and the spread of resistance, but, as of yet, their ecological and clinical implications are unclear.

To address this knowledge gap, we performed a broader screen for bacteriocinogenic *E. coli* strains in fecal samples from healthy individuals, identifying isolates that exhibited antimicrobial activity against *E. coli* DH5α. The main objective of this study was to investigate the genomic features of bacteriocin-producing *E. coli* from the gut microbiota of non-clinical subjects, with a particular focus on the co-occurrence of bacteriocin genes, ARGs, and virulence factors. In doing so, we sought to assess their potential implications for gut microbial ecology and the dissemination of antibiotic resistance.

## 2. Results

We screened 3107 colonies from 109 fecal samples of healthy individuals in the Singapore Integrative Omics Study for antimicrobial activity against *E. coli* DH5α using a large-scale spot-on-lawn assay (Figure 1). Colonies were initially selected on MacConkey agar to enrich for gram-negative bacteria. One representative inhibitory isolate was chosen per positive sample. The previously described MDR strain 94EC was included due to its clinical relevance. In total, 38 *E. coli* isolates were selected for whole-genome comparative analysis. The presence of bacteriocin genes, antibiotic resistance genes, and virulence factors was predicted as described in the Section 4.

The *E. coli* isolates (Figure 2) were predicted to belong to five distinct phylogroups. The detected phylogroups were A (15/38, 39.5%) and D (8/38, 21.1%), followed by B1 (6/38, 15.8%), B2 (5/38, 13.2%), and then F (4/38, 10.5%). All 38 isolates were assigned to 23 unique sequence types (STs) and one unassigned (D96). Of note, ST10 was the most prevalent (8 isolates), followed by ST69 (4 isolates); among the remaining STs, five were doubletons and 17 were singletons (including one unassigned). The core genome phylogenetic tree illustrates the distribution of phylogroups and STs. Specifically, phylogroups B2 and F fall into a distinct group (Group I) comprising nine isolates belonging to seven unique STs while A, B1 and D make up another group (Group II) consisting of 29 isolates belonging to 17 unique STs.

Bacteriocin gene cluster prediction revealed a vast spread of bacteriocin types among 35 out of the 38 isolates (92%). A total of 17 bacteriocin variants were identified, of which tenwere colicins, six are microcins and one pesticin. Of the 38 isolates, seven harbored 3–5 bacteriocin variants, 28 harbored 1–2 variants, and three (found exclusively in phylogroup D) did not possess any known bacteriocins despite producing inhibitory activity against *E. coli* DH5α growth. ColE1 was the most frequently occurring bacteriocin, identified in 13 out of the 38 isolates (34.2%). The majority of ColE1 carriers belonged to phylogroups A, B1, and F and were absent from phylogroup B2, and only one out of nine phylogroup D isolates possessed this bacteriocin.

Interestingly, we identified a chromosomal *hcp-et3-4* gene cluster in seven out of the 38 isolates. Although the *hcp-et3-4* gene product is an effector of the Type VI secretion system (T6SS), it contains the Pyocin S (ET3) and the Colicin-DNase (ET4) bacteriocin domains [7]; hence, we classified this gene cluster as a bacteriocin variant for the scope of this study. We found that all isolates harboring *hcp-et3-4* were restricted to Group I—phylogroup B2 had five isolates and phylogroup F had two isolates. Among the other chromosomally encoded gene clusters, we also identified the siderophore-microcins MccM-H47 and MccM-H47–I47. Of note, the MccM-H47–I47 cluster was the most prevalent, and it was exclusively carried by the Group II isolates—phylogroup A had six isolates and phylogroup D had one isolate. The other two clusters were each present in a single isolate.

The ARG profiles of the isolates can provide valuable insights into the gut antibiotic resistance landscape of healthy individuals in Singapore. The prediction of ARG clusters identified resistance to 12 unique classes of antibiotics—the most prevalent being tetracycline and beta-lactam (each 18/38, 47.4%), followed by aminoglycoside (15/38, 39.5%), trimethoprim (12/38, 31.6%), sulfonamide (11/38, 28.9%), and fluoroquinolone (6/38, 15.8%). About one-third of all isolates were predicted to not carry any known ARGs.

Strains are considered multi-drug resistant (MDR) if they are resistant to at least three distinct classes of antibiotics [8]. Our analysis revealed that nearly 40% of isolates (15/38) were MDR. Most prominently, D96 was predicted to be resistant to eight classes; D72, D108, and 94EC were resistant to seven classes; and D82 and D93 were resistant to six classes. A majority of ARGs were located on plasmids (68.7%); 28.4% were located on the chromosome only, and 2.9% were found on both. Interestingly, eight isolates were predicted to have most, if not all, ARGs located on the chromosome, out of which half of them were MDR. This presents an opportunity for investigating the evolution of the intrinsic (chromosomal) resistome, which is necessary for predicting the likelihood of emergence of antibiotic resistance in bacterial populations [9].

In addition to bacteriocin gene and ARG profiling, the diversity of virulence factors (VFs) among these isolates was also elucidated. On average, the Group I isolates carried substantially more VFs than the Group II isolates (Group I: 33.2 ± 6.9 VFs; Group II: 22.9 ± 5.7 VFs). The isolates of phylogroups B2 and F (Group I) have frequently been implicated as extraintestinal pathogenic *E. coli* (ExPEC) in humans, capable of causing infections outside of the gastrointestinal tract. In our dataset, we found that 55.3% (21/38) of the isolates were considered ExPEC (≥2 ExPEC VF markers [10]). A further breakdown revealed that two isolates (D75 and D99) carried four ExPEC markers; seven isolates carried three ExPEC markers; and 12 isolates harbored two ExPEC markers. Indeed, it was observed that all Group I isolates were ExPEC strains and possessed ≥3 ExPEC markers (except D105), whereas slightly less than half of the Group II isolates were ExPEC, and all of these Group II isolates carried ≤2 markers, except D95 (phylogroup D)—which carried 3 markers. ExPEC strains are of clinical significance, as they can cause a myriad of non-intestinal infections in the human body and have the ability to transmit resistance genes to other pathogenic bacteria. Our study showed a relatively high proportion of ExPEC strains present in the feces of healthy human subjects. Furthermore, about one-third of these were MDR, thereby raising concerns of potential problematic infections by these opportunistic pathogens.

## 3. Discussion

Here, we show evidence for a diverse profile of DH5α-inhibiting *E. coli* in a healthy Singaporean cohort. These strains were made up of at least 23 unique STs belonging to five phylogroups, with phylogroup A being the most prevalent. Our findings contrast with a previous study conducted on clinical samples that reported a higher prevalence in phylogroup B2 *E. coli* [8]. A possible reason for this observation could be the difference in sample sources—isolates in this study were derived from fecal samples of healthy subjects, whereas Šmajs et al. analyzed isolates from urinary tract infections (UTIs). Phylogroup B2 *E. coli* are often associated with UTIs, and Mcc-H47 has been suggested to be an important determinant for facilitating the colonization and subsequent emergence of phylogroup B2 strains from the intestinal reservoir [9,10]. Hence, this may explain the higher incidence of Mcc-H47 observed in clinical UTI samples compared with our dataset.

Most of the obtained isolates (92%) carried bacteriocins. Our findings are concordant with the fact that colicins are typically plasmid encoded, while microcins can be either plasmid or chromosomally encoded [7]. Furthermore, a large proportion of them were MDR and ExPEC strains based on bioinformatic prediction. Approximately 70% of the detected ARGs were located on plasmids that could then act as molecular vehicles to spread resistance genes to highly virulent pathogens, thus posing a threat to treatment using conventional antibiotics. Further experimental research will be necessary to investigate the in vivo impact of the co-occurrence of bacteriocin genes, ARGs, and virulence factors in these strains. In parallel, it will also be important to assess whether these isolates have the potential to suppress multi-drug-resistant bacteria under in vivo conditions. Such findings would support recent suggestions that bacteriocins could serve as viable antimicrobial agents [4]. A key advantage lies in the specificity of the bacteriocins that the isolates produce that can selectively target pathogenic bacteria (typically close relatives of the bacteriocin producer); thus, the narrow antimicrobial spectrum minimizes disruption to the surrounding microbiota. For instance, Mortzfeld et al. showed that Mcc-I47 exhibits specific inhibitory activity against Enterobacteriaceae strains, and its potency is comparable to commonly prescribed antimicrobials [11]. The targeted antimicrobial activity of bacteriocins, combined with their origin as commensal gut microbes, makes the isolates potentially suitable as probiotic candidates. However, safety and efficacy must be ensured by selecting isolates devoid of ARGs and can also be further optimized by genetic engineering to remove undesirable genomic elements. Lastly, since our analysis was conducted on isolates from healthy individuals, it will be essential to compare these findings with *E. coli* strains isolated from clinical settings to better understand their potential implications for human health.

## 4. Materials and Methods

### 4.1. Isolation and Screening

Fecal samples from 109 individuals from the 2018 Singapore Integrative Omics Study were collected and processed as described previously [1,2,3]. Fecal matter was inoculated in Luria–Bertani (LB) broth for 3 h of shaking at 200 rpm at 37 °C and then plated onto MacConkey agar and incubated overnight at 37 °C. Colonies (*n* = 3107) were picked and spotted onto a lawn of *Escherichia coli* DH5α, which was chosen due to its non-pathogenic and non-bacteriocin-producing qualities. The strain is highly susceptible to bacteriocins produced by other strains. The plate was incubated at 37 °C for 18 h before checking for the zone of inhibition of *Escherichia coli* DH5α. Colonies of *Escherichia coli* DH5α served as negative control, with 94EC serving as positive control. Colonies that had a visible zone of inhibition larger than 0.5–1 mm were selected for further processing (see Appendix A for an example). DNA extraction and whole-genome sequencing were performed for 38 isolates that exhibited inhibitory activity.

### 4.2. DNA Extraction and Sequencing

Genomic DNA was extracted using Qiagen Genomic-tip 20/G according to the manufacturer’s protocol. DNA libraries were prepared using an ONT rapid barcoding kit (SQK-RBK004) and sequenced on MinION R9.4.1 flow cells. Sequenced reads were then basecalled using guppy on high accuracy mode. Genomic DNA was also sequenced on NovaSeq 6000 PE150 by a third-party vendor (NovogeneAIT) to generate short reads. Hybrid genome assembly was performed using Unicycler with default parameters [12]. Complete plasmid sequences were obtained for the majority of isolates with circularized assemblies; where this was not possible, contigs carrying plasmid replicon sequences were used to infer plasmid origin [13].

### 4.3. Bioinformatics

The presence of bacteriocin gene clusters was predicted with a BAGEL4 webserver [14]. Antibiotic resistance and virulence factor gene clusters were predicted using ResFinder v4.5.0 and VirulenceFinder v2.0.5 at 90% identity and coverage with other default parameters [15,16,17]. Isolates were considered ExPEC when ≥2 ExPEC VF markers were detected [10]. Phylogroups and sequence types were determined with EzClermont and MLST v2.0.9 using default parameters [18,19]. The phylogenetic tree was inferred by maximum likelihood using IQ-TREE v2.3.6 supported with 1000 replications approximated by an ultrafast bootstrap on a concatenated multiple sequence alignment of 81 single-copy core genes generated from a UBCG2 pipeline using default parameters [20,21,22]. The heatmap was generated in Python 3 using the pandas, seaborn, and matplotlib modules.

## Figures and Tables

**Figure 1 antibiotics-14-00860-f001:**
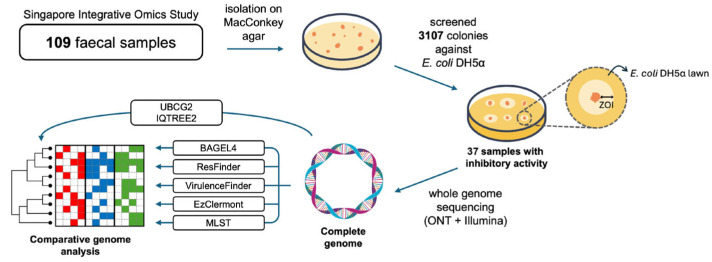
Schematic overview of workflow: fecal samples were screened on MacConkey agar and then against *E. coli* DH5α for inhibitory activity. Whole-genome sequencing and comparative genomic analysis were carried out for positive isolates.

**Figure 2 antibiotics-14-00860-f002:**
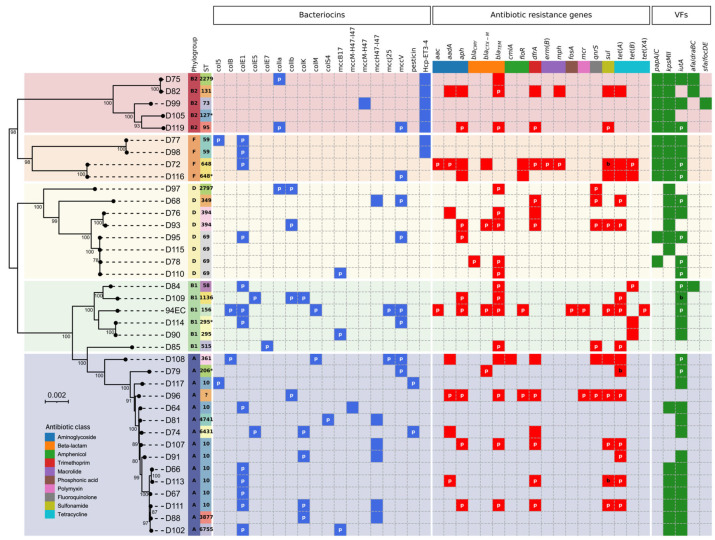
Heatmap illustrating distribution of genomic factors among *E. coli* isolates. Presence of bacteriocin, antibiotic resistance, and virulence factor gene clusters are marked by colored tiles of blue, red, and green, respectively. Non-solid tiles indicate absence of genes. Genomic location of genes is represented by letters: none—chromosome, p—plasmid, and b—both chromosome and plasmid. Maximum likelihood phylogenetic tree was constructed using concatenated alignment of 81 core genes supported by 1000 ultrafast bootstrap replicates. Only bootstrap values ≥ 70% are shown. Scale bar: 0.002 nucleotide substitutions per site.

## Data Availability

Data of sequenced genomes are available under NCBI BioProject ID PRJNA1258756.

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
