# Peer review of "Comparative Genomics of DH5α-Inhibiting Escherichia coli Isolates from Feces of Healthy Individuals Reveals Common Co-Occurrence of Bacteriocin Genes with Virulence Factors and Antibiotic Resistance Genes"

_antibiotics, 2025, doi:10.3390/antibiotics14090860_

Round 1
Reviewer 1 Report
Comments and Suggestions for Authors
The present manuscript studied the comparative genomics of DH5α-inhibiting Escherichia coli isolates from human feces reveals common co-occurrence of bacteriocin genes with virulence factors and antibiotic resistance genes. The topic is interesting, but in my opinion, the following issues should be addressed:
- The authors said that they screened 3107 colonies against common laboratory Escherichia coli strain DH5 in a large-scale spot-on-lawn assay. It is better to show some representative inhibition figures. Also for the antibiotic resistance, I think it is better to perform the antibiotic resistance assay to verify the bioinformatic information.
- Why DH5a was used as the suppressed bacterium? The authors should give brief description in the Material and method section, or the result section.
- In the introduction section, certain statements should be supported by references. For example, the statement in line 57-59.
- According to my question 1, I think some important experiments shoul be done and the Material and method section should be more. For example, inhibition assay, antibiotic resistance, etc.
Author Response
The present manuscript studied the comparative genomics of DH5α-inhibiting Escherichia coli isolates from human feces reveals common co-occurrence of bacteriocin genes with virulence factors and antibiotic resistance genes. The topic is interesting, but in my opinion, the following issues should be addressed:
R1.C1: The authors said that they screened 3107 colonies against common laboratory Escherichia coli strain DH5 in a large-scale spot-on-lawn assay. It is better to show some representative inhibition figures. Also for the antibiotic resistance, I think it is better to perform the antibiotic resistance assay to verify the bioinformatic information.
Thank you, we have included a representative image as a references (see supporting figure 1). We have also expanded and mostly rewritten the text in the Materials and Methods (Line 199-211) to more explicitly describe the spot-on-lawn procedure, including incubation conditions and scoring criteria, to allow readers to understand the assay format.
We agree that antibiotic resistance assays could further verify the bioinformatic information. However, most strains harbor resistance genes to multiple resistance genes, often more than five, which would require a large number of assays to verify the function of genes that share high sequence identity with already functionally verified genes. We therefore think that such assays are beyond scope for this article format (brief report).
R1.C2: Why DH5a was used as the suppressed bacterium? The authors should give brief description in the Material and method section, or the result section.
We have added a brief explanation in the Materials and Methods to clarify that DH5α was selected as the indicator strain because it is a well-characterized, non-pathogenic laboratory E. coli strain that is highly susceptible to bacteriocins, making it a suitable and reproducible target for screening inhibitory activity.
Line 199-211: “Fecal samples from 109 individuals from the 2018 Singapore Integrative Omics Study were collected and processed as described previously [1-3]. Fecal matter was inoculated in Luria-Bertani (LB) broth for 3 hours shaking at 200 rpm, 37°C, then plated onto MacConkey agar and incubated overnight at 37°C. Colonies (n=3107) were picked and spotted onto a lawn of Escherichia coli DH5α, which was chosen as it is non-pathogenic and also non-bacteriocin producing. The strain is highly susceptible to bacteriocins produced by other strains. The plate was incubated at 37°C for 18 hours before checking for zone of inhibition of Escherichia coli DH5α. Colonies of Escherichia coli DH5α served as negative control, 94EC as positive control. Colonies that have a visible zone of inhibition larger than 0.5-1mm were selected for further processing (see supporting figure S1 for an example). DNA extraction and whole genome sequencing were performed for 38 isolates that exhibited inhibitory activity.”
R1.C3: In the introduction section, certain statements should be supported by references. For example, the statement in line 57-59.
Thank you for the comment. We have added the following three references in the introduction.
Cotter, P.D.; Ross, R.P.; Hill, C. Bacteriocins—a viable alternative to antibiotics? Nature Reviews Microbiology 2013, 11, 95-105.
Heilbronner, S.; Krismer, B.; Brötz-Oesterhelt, H.; Peschel, A. The microbiome-shaping roles of bacteriocins. Nature Reviews Microbiology 2021, 19, 726-739.
Riley, M.A.; Wertz, J.E. Bacteriocins: evolution, ecology, and application. Annual Reviews in Microbiology 2002, 56, 117-137.
R1.C4: According to my question 1, I think some important experiments shoul be done and the Material and method section should be more. For example, inhibition assay, antibiotic resistance, etc.
We thank the reviewer for this comment and have also partially addressed this in the response to C1. We would like to emphasize again -as outlined in the title- we analysed the co-occurrence of bacteriocins with virulence factors and antibiotic resistance genes. Any further experimental verification of the antimicrobial activity of all these genes is beyond scope for the article format.
We have strengthened the methodological descriptions and added clarifying statements in the Discussion to acknowledge this limitation and to point to future work that could address it.
Line 175: Furthermore, a large proportion of them were MDR and ExPEC strains as determined by bioinformatic prediction.
Line 179-181 Further experimental research will be necessary to investigate the in vivo impact of the co-occurrence of bacteriocins, ARGs, and virulence factors in these strains.
Reviewer 2 Report
Comments and Suggestions for Authors
Bacterial multidrug resistance (MDR) has become a public health security issue, which may be due to the dissemination of antibiotic resistance genes (ARGs) among the bacteria. This manuscript focused on distribution of the gene clusters associated with the bacteriocin, antibiotic resistance, and virulence factors in the E. coli population isolated from the healthy individual feces in Singapore. They found 17 bacteriocin variants among the 38 sequenced E. coli isolates, which belong to five distinct phylogroups. And ~40% (18/38) of the examined E. coli isolates were MDR, while the ARGs are mainly distributed in plasmids. In addition, the diversity of virulence factors among these isolates was also elucidated. Anyway, this work revealed the co-occurrence of bacteriocins, ARGs, and virulence genes in the gut-residing E. coli, which may remodulate gut microbial dynamics and antibiotic resistance.
However, some issues should be addressed or concerned by the authors.
- The authors said they sequenced 48 isolates in M & M, but only 38 were reported in the results. How about the other ten isolates?
- In Line 126, about 40% of isolates (15/38) are assigned as MDR by the authors. Did the authors determine the antibiotic resistance phenotype of the 38 isolates experimentally? The resistance phenotypes may not always be consistent with the occurrence of ARGs. Therefore, it is necessary to determine the real resistance phenotypes against antibiotics.
- A Veen diagram could be introduced to illustrate the co-occurrence of bacteriocins, ARGs, and virulence factors among the coli isolates used in this manuscript.
- The authors are encouraged to discuss or compare the distribution patterns of ARGs and bacteriocins in their dataset with those from other studies, in order to ascertain whether there exist any differences among various populations, particularly between those from healthy individuals and clinical settings.
Author Response
Comments and Suggestions for Authors
Bacterial multidrug resistance (MDR) has become a public health security issue, which may be due to the dissemination of antibiotic resistance genes (ARGs) among the bacteria. This manuscript focused on distribution of the gene clusters associated with the bacteriocin, antibiotic resistance, and virulence factors in the E. coli population isolated from the healthy individual feces in Singapore. They found 17 bacteriocin variants among the 38 sequenced E. coli isolates, which belong to five distinct phylogroups. And ~40% (18/38) of the examined E. coli isolates were MDR, while the ARGs are mainly distributed in plasmids. In addition, the diversity of virulence factors among these isolates was also elucidated. Anyway, this work revealed the co-occurrence of bacteriocins, ARGs, and virulence genes in the gut-residing E. coli, which may remodulate gut microbial dynamics and antibiotic resistance.
We thank the reviewer for the thoughtful summary of our study and for the constructive comments and suggestions, which have helped us to improve the clarity and completeness of the manuscript. We address each point below:
However, some issues should be addressed or concerned by the authors.
R2.C1 The authors said they sequenced 48 isolates in M & M, but only 38 were reported in the results. How about the other ten isolates?
We appreciate the reviewer’s attention to this detail and apologize for this error. We have corrected the numbering throughout the manuscript.
37 isolates were obtained from the screen of the colonies.
One additional isolate was included from a previous study; hence comparative analysis was performed on 38 isolates in total (37 isolates from screen + 94EC from a previous study).
R2.C2. In Line 126, about 40% of isolates (15/38) are assigned as MDR by the authors. Did the authors determine the antibiotic resistance phenotype of the 38 isolates experimentally? The resistance phenotypes may not always be consistent with the occurrence of ARGs. Therefore, it is necessary to determine the real resistance phenotypes against antibiotics.
We thank the reviewer for this comment. However, we would like to emphasize -as outlined in the title- we analysed the co-occurrence of bacteriocins with virulence factors and antibiotic resistance genes. Any further experimental verification of the antimicrobial activity of all these genes is beyond scope for the article format.
We agree that antibiotic resistance assays could further verify the bioinformatic information. However, most strains harbor resistance genes to multiple resistance genes, often more than five, which would require a large number of assays to verify the function of genes that share high sequence identity with already functionally verified genes. We therefore think that such assays are beyond scope for this article format (brief report).
We have added the following sentences to the discussion to acknowledge this limitation:
Line 175: Furthermore, a large proportion of them were MDR and ExPEC strains as determined by bioinformatic prediction.
Line 179-181 Further experimental research will be necessary to investigate the in vivo impact of the co-occurrence of bacteriocins, ARGs, and virulence factors in these strains.
R2.C3: A Veen diagram could be introduced to illustrate the co-occurrence of bacteriocins, ARGs, and virulence factors among the coli isolates used in this manuscript.
While this could be an interesting suggestions, we may need further clarification from the reviewer regarding the outline of such a diagram. As shown in Figure 2, there appear no core bacteriocins, different combinations of ARGs and VF are conserved across all strains. We have therefore chosen the present outline to visualize the unique combinations present in each strain.
R2.C4: The authors are encouraged to discuss or compare the distribution patterns of ARGs and bacteriocins in their dataset with those from other studies, in order to ascertain whether there exist any differences among various populations, particularly between those from healthy individuals and clinical settings.
We thank the reviewer for this interesting suggestion and may consider this in a future expansion of the analyses. However, such work would be beyond scope for this article format (brief report: short, observational studies that report preliminary results or a short complete study or protocol).
We have added a sentence in the discussion that the proposed analyses should be performed in the future.
Line 193: Lastly, since our analysis was conducted on isolates from healthy individuals, it will be essential to compare these findings with E. coli strains isolated from clinical settings to better understand their potential implications for human health.
Reviewer 3 Report
Comments and Suggestions for Authors
In this manuscript, the authors screened Escherichia coli strains isolated from the feces of healthy human individuals in Singapore for antimicrobial activity against E. coli DH5α. A total of 37 representative strains were subjected to whole-genome sequencing and comparative genomic analysis, revealing five phylogroups and 23 unique sequence types. The authors compared the types and distributions of bacteriocin gene clusters, antibiotic resistance genes, and virulence-associated genes among these strains. Interestingly, a relatively high proportion of the isolates fell into the category of extraintestinal pathogenic E. coli (ExPEC), despite being derived from healthy individuals. The manuscript is generally well-written and provides valuable data to the field of antimicrobial research. However, I have several comments that should be addressed.
- Title: The title should indicate that the E. coli isolates were derived from “healthy” individuals, as this is an important feature of the study.
- Antimicrobial Activity Assay: The authors should provide more detailed information on how antimicrobial activity was assessed. For example, what was the threshold zone of inhibition diameter used to define antimicrobial activity? What were the positive and negative controls used in the assay?
- Lines 69–70: The authors refer to “supporting information”, but I could not find any such material included with the manuscript.
- Figure 2: The manuscript shows whether genes of interest are located on chromosomes and/or plasmids. However, the methods used to determine the genomic location (i.e., plasmid vs. chromosomal) are not described in sufficient detail. The authors should elaborate on how plasmid-borne genes were identified. Did the authors assemble complete plasmid sequences for each strain? The potential mobility of plasmid-encoded genes is an important point, especially if mobilization elements are present. If possible, the authors should also compare plasmid types across the strains.
- Discussion Section: The discussion is brief, and the proposed application of bacteriocin-producing strains as probiotics may be overstated given the data. The authors might consider combining it with the Results section.
Author Response
In this manuscript, the authors screened Escherichia coli strains isolated from the feces of healthy human individuals in Singapore for antimicrobial activity against E. coli DH5α. A total of 37 representative strains were subjected to whole-genome sequencing and comparative genomic analysis, revealing five phylogroups and 23 unique sequence types. The authors compared the types and distributions of bacteriocin gene clusters, antibiotic resistance genes, and virulence-associated genes among these strains. Interestingly, a relatively high proportion of the isolates fell into the category of extraintestinal pathogenic E. coli (ExPEC), despite being derived from healthy individuals. The manuscript is generally well-written and provides valuable data to the field of antimicrobial research. However, I have several comments that should be addressed.
We thank the reviewer for the clear and constructive feedback, as well as for recognising the value of our dataset to the field. We address each point below:
R3.C1:Title: The title should indicate that the E. coli isolates were derived from “healthy” individuals, as this is an important feature of the study.
Thank you, we agree that this is an important distinction and have revised the title to read:
Comparative genomics of DH5α-inhibiting Escherichia coli isolates from the feces of healthy individuals reveals common co-occurrence of bacteriocin genes with virulence factors and antibiotic resistance genes.
R3.C2: Antimicrobial Activity Assay: The authors should provide more detailed information on how antimicrobial activity was assessed. For example, what was the threshold zone of inhibition diameter used to define antimicrobial activity? What were the positive and negative controls used in the assay?
We have expanded the Materials and Methods section to include a more detailed description of the spot-on-lawn assay. Specifically, we now describe the size of inhibition zones used to define positive activity, as well as the use of E. coli 94EC as a positive control and DH5α without bacteriocin production as a negative control.
Lines 203-211:
Colonies (n=3107) were picked and spotted onto a lawn of Escherichia coli DH5α, which was chosen as it is non-pathogenic and also non-bacteriocin producing. The strain is highly susceptible to bacteriocins produced by other strains. The plate was incubated at 37°C for 18 hours before checking for zone of inhibition of Escherichia coli DH5α. Colonies of Escherichia coli DH5α served as negative control, 94EC as positive control. Colonies that have a visible zone of inhibition larger than 0.5-1mm were selected for further processing (see supporting figure S1 for an example). DNA extraction and whole genome sequencing were performed for 38 isolates that exhibited inhibitory activity.
R3.C3: Lines 69–70: The authors refer to “supporting information”, but I could not find any such material included with the manuscript.
Thank you, this has been deleted. The reference to “supporting information” was a remnant from an earlier draft and has now been removed.
R3.C4: Figure 2: The manuscript shows whether genes of interest are located on chromosomes and/or plasmids. However, the methods used to determine the genomic location (i.e., plasmid vs. chromosomal) are not described in sufficient detail. The authors should elaborate on how plasmid-borne genes were identified. Did the authors assemble complete plasmid sequences for each strain? The potential mobility of plasmid-encoded genes is an important point, especially if mobilization elements are present. If possible, the authors should also compare plasmid types across the strains.
Thank you, we have expanded the Materials and Methods section to describe that plasmid-encoded genes were identified using PlasmidFinder and by inspecting hybrid assemblies generated by Unicycler, which integrates short- and long-read sequencing. The suggestion of the reviewer regarding the detailed plasmid typing is of interest but will be beyond scope for this article format.
For clarification we have added the following lines:
Line 218-221: Complete plasmid sequences were obtained for the majority of isolates with circularised assemblies; where this was not possible, contigs carrying plasmid replicon sequences were used to infer plasmid origin [13].
R3.C5: Discussion Section: The discussion is brief, and the proposed application of bacteriocin-producing strains as probiotics may be overstated given the data. The authors might consider combining it with the Results section.
Thank you, we have retained the separation of Results and Discussion to maintain clarity for readers, but have rewritten and extended the Discussion to provide more context for our findings. We have also rephrased the sentence regarding the use of the strains as probiotic.
We hope these revisions address the reviewer’s comments while keeping the study within its intended scope of a Brief Report.
Lines 162-196 (rewritten discussion)
Here, we show evidence for a diverse profile of DH5α-inhibiting E. coli in a healthy Singaporean cohort. These strains were made up of at least 23 unique STs belonging to five phylogroups, with phylogroup A being most prevalent. Our findings contrast with a previous study conducted on clinical samples which reported a higher prevalence in phylogroup B2 E. coli [8]. A possible reason for this observation could be the difference in sample sources - isolates in this study were derived from fecal samples of healthy subjects whereas Šmajs et al. analyzed isolates from urinary tract infections (UTIs). Phylogroup B2 E. coli are often associated with UTIs, and MccH47 has been suggested to be an important determinant for facilitating colonization and subsequent emergence of phylogroup B2 strains from the intestinal reservoir [9,10]. Hence, this may explain the higher incidence of MccH47 observed in clinical UTI samples compared to our dataset.
Most of the obtained isolates (92%) carried bacteriocins. Our findings are concordant with the fact that colicins are typically plasmid-encoded while microcins can be either plasmid- or chromosomally encoded [7]. Furthermore, a large proportion of them were MDR and ExPEC strains based on bioinformatic prediction. Approximately 70% of de-tected ARGs were located on plasmids which may then act as molecular vehicles to spread resistance genes to highly virulent pathogens, thus, posing a threat to treatment by conventional antibiotics. Further experimental research will be necessary to investigate the in vivo impact of the co-occurrence of bacteriocins, ARGs, and virulence factors in these strains. In parallel, it will also be important to assess whether these isolates have the po-tential to suppress multi-drug-resistant bacteria under in vivo conditions. Such findings would support recent suggestions that bacteriocins could serve as viable antimicrobial agents [4]. A key advantage lies in the specificity of the bacteriocins they produce which can selectively target pathogenic bacteria (typically close relatives of the bacteriocin pro-ducer), thus, the narrow antimicrobial spectrum minimizes disruption to the surrounding microbiota. For instance, Mortzfeld et al. showed that MccI47 exhibits specific inhibitory activity against Enterobacteriaceae strains and its potency is comparable to commonly prescribed antimicrobials [11]. The targeted antimicrobial activity of bacteriocins, com-bined with their origin as commensal gut microbes, makes the isolates potentially suitable as probiotic candidates. However, safety and efficacy must be ensured by selecting isolates devoid of ARGs and can also be further optimized by genetic engineering to remove unde-sirable genomic elements. Lastly, since our analysis was conducted on isolates from healthy individuals, it will be essential to compare these findings with E. coli strains iso-lated from clinical settings to better understand their potential implications for human health.
Reviewer 4 Report
Comments and Suggestions for Authors
This brief report presents a comparative genomic analysis of E. coli strains from human faecal samples in Singapore, focusing on bacteriocin production, antibiotic resistance, and virulence factors. The study is relevant due to rising concerns about the gut resistome in healthy populations. However, the study lacks clear justification and conclusion.
Authors should please clarify how many isolates were sequenced in this study, 37 or 38, because of the discrepancy in lines 26 and 76.
Line 20: Please clearly state the objective of this study. What's the rationale?
Line 27: Please insert the % of isolates that belonged to each phylogroup.
Line 28: What % of the isolates had bacteriocin genes?
Line 31: How many isolates had resistance genes against up to eight classes of antibiotics?
Lines 33-34: "potential extraintestinal pathogenic E. coli (ExPEC)"...what criteria was used for this classification.
Lines 68-80: These lines are supposed to be in the Materials & Methods section. Please revist these lines and remove parts that are already stated in the Materials & Methods section.
Lines 93-94: Please move this sentence to the discussion section.
Lines 110-118: Please move these sentences to the discussion section.
Lines 138-139: Please move these sentences to the discussion section.
Line 224: Please ensure that readers can access the genome sequences
Author Response
This brief report presents a comparative genomic analysis of E. coli strains from human faecal samples in Singapore, focusing on bacteriocin production, antibiotic resistance, and virulence factors. The study is relevant due to rising concerns about the gut resistome in healthy populations. However, the study lacks clear justification and conclusion.
We thank the reviewer for their thoughtful comments and suggestions, which have helped to clarify and strengthen the manuscript. Below, we address each point in turn:
R4.C1: Authors should please clarify how many isolates were sequenced in this study, 37 or 38, because of the discrepancy in lines 26 and 76.
Number of sequenced isolates
Thank you, we apologize for the mistake. This has been corrected throughout.
37 isolates were obtained from the screen of the colonies.
One additional isolate was included from a previous study, hence comparative analysis was performed on 38 isolates in total (37 isolates from screen + 94EC from previous study).
R4.C2: Line 20: Please clearly state the objective of this study. What's the rationale?
Thank you! We have largely restructured the introduction to emphasis more the rationale and objective.
Main changes have been made to lines 58-72:
Genome analysis revealed the presence of several bacteriocin gene clusters in the strain (previously unpublished). Bacteriocins are known to have important roles in ecolo-gy and microbial population dynamics and have also been of interest for applications in biotechnological processes or as potential alternatives for antibiotic therapeutics [4-6]. However, their co-occurrence with antibiotic resistance genes and their role for the spread of antibiotic resistance or in stabilizing the resistome are less well understood. Such com-binations may influence microbial fitness, strain persistence, and the spread of resistance, yet their ecological and clinical implications are unclear.
To address this knowledge gap, we performed a broader screen for bacteriocinogenic E. coli strains in fecal samples from healthy individuals, identifying isolates that exhibited antimicrobial activity against E. coli DH5α. The main objective of this study was to inves-tigate the genomic features of bacteriocin-producing E. coli from the gut microbiota of non-clinical subjects, with a particular focus on the co-occurrence of bacteriocin genes, ARGs, and virulence factors. In doing so, we sought to assess their potential implications for gut microbial ecology and the dissemination of antibiotic resistance
R4.C3: Line 27: Please insert the % of isolates that belonged to each phylogroup. Percentages for phylogroups
Thank you, the percent isolates that belonged to each phylogroup are shown in line 84-85 of the manuscript:
The E. coli isolates (Figure 2) were predicted to belong to five distinct phylogroups. The most dominant phylogroups are A (15/38, 39.5%) and D (8/39, 21.1%), followed by B1 (6/38, 15.8%), B2 (5/38, 13.2%) then F (4/38, 10.5%).
R4.C4: Line 28: What % of the isolates had bacteriocin genes?
Thank you, we have added this information to the Abstract and Results:
Line 28: Bacteriocin gene clusters were widespread (92% of isolates carried one or more bacteriocin gene clusters),
Line 93-94: Bacteriocin gene cluster prediction reveals a vast spread of bacteriocin types among 35 out of 38 isolates (92%).
R4.C5: Line 31: How many isolates had resistance genes against up to eight classes of antibiotics?
We were not entirely clear about the reviewer comment as the sentence in line 31-32 already states that 1 strain encodes for 8 different classes of antibiotics. We have specified the sentence so that the strain ID is also mentioned.
Line 31-32: Approximately 40% of the sequenced isolates were MDR, with resistance for up to eight antibiotic classes in one strain (strain D96).
R4.C6: Lines 33-34: "potential extraintestinal pathogenic E. coli (ExPEC)"...what criteria was used for this classification.
We have clarified in the Materials and Methods that ExPEC classification was based on the presence of two or more of the five virulence markers described by Johnson et al. (2003).
Line 230: [14]. Antibiotic resistance and virulence genes were predicted using ResFinder v4.5.0 and VirulenceFinder v2.0.5 at 90% identity and coverage with other parameters default [15-17]. Isolates were considered ExPEC when ≥2 ExPEC VF markers were detected [10].
R4.C7: Lines 68-80: These lines are supposed to be in the Materials & Methods section. Please revist these lines and remove parts that are already stated in the Materials & Methods section.
Thank you. The respective lines in the opening paragraph have been rewritten and moved to the Materials and Methods.
Please see line 75-82
“We screened 3,107 colonies from 109 fecal samples of healthy individuals in the Sin-gapore Integrative Omics Study for antimicrobial activity against E. coli DH5α using a large-scale spot-on-lawn assay. Colonies were initially selected on MacConkey agar to en-rich for gram-negative bacteria. One representative inhibitory isolate was chosen per posi-tive sample. The previously described MDR strain 94EC was included due to its clinical relevance. In total, 38 E. coli isolates were selected for whole-genome comparative analysis. Presence of bacteriocin, antibiotic resistance and virulence genes were predicted as de-scribed in the Materials and Methods section.”
R4.C8: Lines 93-94: Please move this sentence to the discussion section.
Thank you, this has been moved
R4.C9: Lines 110-118: Please move these sentences to the discussion section.
Thank you, this has been moved
R4.C10: Lines 138-139: Please move these sentences to the discussion section.
Thank you for the suggestion. In this case we decided to keep the sentences in place as they are provide a transition to the ExPEC, which are analysed in more detail in the subsequent lines.
R4.C11: Line 224: Please ensure that readers can access the genome sequences
We have confirmed that all genome sequences are available at NCBI under BioProject ID PRJNA1258756 and have ensured that the accession number is clearly stated in the Data Availability Statement.
Round 2
Reviewer 1 Report
Comments and Suggestions for Authors
The authors have made corresponding revisions, and I have no further comments now.